# Enhancement of Astaxanthin Biosynthesis in Oleaginous Yeast *Yarrowia lipolytica* via Microalgal Pathway

**DOI:** 10.3390/microorganisms7100472

**Published:** 2019-10-19

**Authors:** Larissa Ribeiro Ramos Tramontin, Kanchana Rueksomtawin Kildegaard, Suresh Sudarsan, Irina Borodina

**Affiliations:** The Novo Nordisk Foundation Center for Biosustainability, Technical University of Denmark, Kemitorvet 220, 2800 Kgs. Lyngby, Denmark; lartra@biosustain.dtu.dk (L.R.R.T.); kanchana@biophero.com (K.R.K.); sursud@biosustain.dtu.dk (S.S.)

**Keywords:** *Yarrowia lipolytica*, β-carotene, astaxanthin, metabolic engineering, submerged fermentation

## Abstract

Astaxanthin is a high-value red pigment and antioxidant used by pharmaceutical, cosmetics, and food industries. The astaxanthin produced chemically is costly and is not approved for human consumption due to the presence of by-products. The astaxanthin production by natural microalgae requires large open areas and specialized equipment, the process takes a long time, and results in low titers. Recombinant microbial cell factories can be engineered to produce astaxanthin by fermentation in standard equipment. In this work, an oleaginous yeast *Yarrowia lipolytica* was engineered to produce astaxanthin at high titers in submerged fermentation. First, a platform strain was created with an optimised pathway towards β-carotene. The platform strain produced 331 ± 66 mg/L of β-carotene in small-scale cultivation, with the cellular content of 2.25% of dry cell weight. Next, the genes encoding β-ketolase and β-hydroxylase of bacterial (*Paracoccus sp*. and *Pantoea ananatis*) and algal (*Haematococcus pluvialis*) origins were introduced into the platform strain in different copy numbers. The resulting strains were screened for astaxanthin production, and the best strain, containing algal β-ketolase and β-hydroxylase, resulted in astaxanthin titer of 44 ± 1 mg/L. The same strain was cultivated in controlled bioreactors, and a titer of 285 ± 19 mg/L of astaxanthin was obtained after seven days of fermentation on complex medium with glucose. Our study shows the potential of *Y. lipolytica* as the cell factory for astaxanthin production.

## 1. Introduction

Astaxanthin is a keto-carotenoid compound with a red color and strong antioxidant activity. It is mainly used in aquaculture as a feed additive and in human nutrition as a dietary supplement [1]. Astaxanthin and the closely related compound canthaxanthin are also used in the diet of quails and chicken for a more intense color of the flesh and egg yolks [2,3]. Astaxanthin is produced by chemical synthesis (ca. 90%) and by algal fermentation. The chemically synthesized astaxanthin is not allowed for human consumption due to the presence of several chiral forms of astaxanthin as well as some other impurities. Therefore, astaxanthin is mainly used in aquafarming [4,5]. Astaxanthin market price varies from $2500–7000/kg and comprises a significant fraction of the salmon production cost (up to 15%) [6]. The natural astaxanthin is primarily extracted from the freshwater green alga *Haematococcus pluvialis*, which can accumulate 1.5–3% astaxanthin on a dry cell weight (DCW) basis and is the richest source for natural production of astaxanthin [7,8]. The astaxanthin chemical structure varies between three different stereoisomers, (3S, 3′S), (3R, 3′S), and (3R, 3′R). In chemically synthetized astaxanthin, these isomers are obtained in the ratio of 1:2:1. The most valuable one is the 3S, 3′S stereoisomer, which is predominantly found in *H. pluvialis* [9]. To address the high demand for astaxanthin, efforts have been made to increase the astaxanthin production in the natural producer organisms, such as *H. pluvialis* and red yeast *Xanthophyllomyces dendrorhous*, by metabolic engineering. Studies made by Gassel et al. (2013) used random mutagenesis, overexpression of the bifunctional phytoene synthase/lycopene cyclase (*crtYB*), and astaxanthin synthase (*asy*), and selection of an optimum growth medium to reach an astaxanthin content of 9.7 mg/g DCW by *X. dendrorhous* in fermenters [10]. In another work, Gassel et al. (2014) reported an astaxanthin content of 9 mg/g DCW by *X. dendrorhous*, which was obtained in shake-flask culture after a combination of classical mutagenesis and simultaneous integration of rate-limiting enzymes encoded by genes from *X. dendrorhous* (*crtYB*, *asy*, geranylgeranyl pyrophosphate synthase (*crtE*), and 3-hydroxy-3-methylglutaryl-coenzymeA reductase (*HMG*) in its truncated form lacking the membrane binding region) [11]. The production of astaxanthin by the microalgae *H. pluvialis* has been improved primarily by classical mutagenesis and selection [9]. In 2006, a transformation protocol for this microalga was reported by Steinbrenner and Sandmann, where they transformed *H. pluvialis* with a mutated phytoene desaturase (*PDS* gene) and obtained a transformant with 32% higher astaxanthin content than the wild type. In shake flask cultivation, this strain accumulated 11.4 mg/g DCW astaxanthin [12]. Recently, new approaches such as nuclear transformation vectors [13] and genetic engineering of chloroplasts genome [14] have been developed. However, a feasible natural production of astaxanthin by *H. pluvialis*, able to compete with chemical synthesis, was not yet achieved. The astaxanthin biosynthesis has also been engineered into noncarotenogenic organisms, such as the bacterium *Escherichia coli*, and yeasts *Saccharomyces cerevisiae* and *Yarrowia lipolytica*. Park et al. (2018) used *E. coli* as platform for production of astaxanthin by expressing heterologous genes *crt* (*crtE*, lycopene cyclase (*crtY*), phytoene desaturase (*crtI*), phytoene synthase (*crtB*), and β-carotene hydroxylase (*crtZ*)) from *Pantoea ananatis* and a truncated β-carotene ketolase gene (*trCrBKT*) from *Chlamydomonas reinhardtii*. The authors used the signal peptide of OmpF and TrxA to tag the N-terminus and C-terminus of *trCrBKT* and confer stable expression and to efficiently guide *trCrBKT* to the *E. coli* membrane. Further optimization of culture conditions and overexpression of 2-C-methyl-D-erythritol 4-phosphate cytidylyltransferase (*ispD*) and 4-diphosphocytidyl-2-C-methyl-D-ery-thritol kinase (*ispF*) from *E. coli* lead to an astaxanthin production of 432 mg/l with 7 mg/g DCW in fed-batch fermentation [15]. Another work done in *E. coli* reported production of 320 mg/L of astaxanthin by simultaneous fermentation and extraction using pathway optimization on transcriptional, translational, and enzyme levels. In this approach, Zhang et al. (2018) expressed 14 genes stepwise (grouped into four major modules) to optimize the production of precursors lycopene and β-carotene, and to reduce the bottlenecks towards the production of astaxanthin [16]. Metabolic engineering done in the yeast *S. cerevisiae* compared the activity of the β-carotene hydroxylase (CrtZ) from *Alcaligenes sp.* and CrtZ from *Agrobacterium aurantiacum* towards the production of astaxanthin. In this study, Jin et al. (2018) integrated heterologous genes (β-carotene ketolase (*crtW*) from *Brevundimonas vesicularis* and *crtZ* from *Agrobacterium aurantiacum*), and used mutagenesis by atmospheric and room temperature plasma to promote astaxanthin production. After fermentation in 5 L bioreactor, they obtained an astaxanthin content of 13.8 mg/g DCW (217.9 mg/L) [17]. Studies done by Zhou et al. (2017) improved the pathway towards astaxanthin precursors in *S. cerevisiae* by integrating the gene *crtE* and the rate-limiting enzymes *crtI*, *crtYB* (from *X. dendrorhous*), and truncated *HMG1* (from *S. cerevisiae*). After expressing *OBKTM* and *OCRTZ* (β-carotene ketolase and β-carotene hydroxylase from *H. pluvialis*, respectively, which were developed by directed evolution), the engineered strain accumulated 8 mg/g DCW (47 mg/l) of (3S, 3′S)-astaxanthin in shake-flask cultures [18]. In another work, *S. cerevisiae* was engineered to produce astaxanthin by expression of *crtZ* and *BKT* from *H. pluvialis*. In this work, a content of 4.7 mg/g DCW of astaxanthin was achieved in the shake-flask cultures [19]. A promising organism for production of a variety of carotenoids, including astaxanthin, is the oleaginous yeast *Y. lipolytica*. Due to its biosafety record and for the natural production of carotenoid precursors, cytosolic acetyl-CoA, and redox co-factor NADPH, *Y. lipolytica* has the potential to produce astaxanthin at high titers [20,21,22]. Our group reported engineering of *Y. lipolytica* for astaxanthin production in Kildegaard et al. (2017). After metabolic engineering and integration of several heterologous genes for the production of astaxanthin, we obtained 54.6 mg/L (3.5 mg/g DCW) of astaxanthin. To achieve this titer, we expressed the bifunctional phytoene synthase/lycopene cyclase (*crtYB*) and the phytoene desaturase (*crtI*) from *X. dendrorhous*. We further optimized and expressed *HMG1* and compared the activity of the geranylgeranyl diphosphate synthases *GGS1* and *crtE* from *Y. lipolytica* and *X. dendrorhous*, respectively. Next, we downregulated the competing squalene synthase *SQS1* by truncating the native promoter to 50 bp and introduced the astaxanthin pathway by expressing and optimizing the copy numbers of the β-carotene ketolase (*crtW*) from *Paracoccus sp*. and the β-carotene hydroxylase (*crtZ*) from *Pantoea ananatis* [23]. A list of the astaxanthin content produced by the above-mentioned organisms and their genotype is presented in Table 1. In this present work, we aimed to produce a high astaxanthin producing *Y. lipolytica* strain. To increase the production of the precursor β-carotene, we expressed and compared two different geranylgeranyl pyrophosphate synthases (CrtE and GGPPs7). The best β-carotene producer strain was used as the platform for integration of heterologous astaxanthin genes (*crtW*, *BKT*, and two different *crtZ*) in different combinations. To optimize the astaxanthin production, different molar ratios of the astaxanthin genes were tested. The best performing astaxanthin producer strain was cultivated in controlled bioreactors.

## 2. Results

### 2.1. Enhancement of Beta-Carotene Production by the Introduction of crtE and GGPPs7

Astaxanthin is biologically synthesized from β-carotene, which in turn is made from two products of the mevalonate pathway, isopentenyl pyrophosphate (IPP) and dimethylallyl pyrophosphate (DMAPP). An efficient strategy to enhance IPP and DMAPP production in *Y. lipolytica* is the upregulation of the native mevalonate pathway genes and downregulation of side fluxes, e.g., towards squalene [23]. For biosynthesis of β-carotene in *Y. lipolytica*, insertion of the heterologous β-carotene pathway genes (*crtE*, *crtYB*, and *crtI*) is required. To improve the carbon flux towards β-carotene biosynthesis, we chose to enhance the conversion of FPP into GGPP by inserting two different GGPP synthases, encoded by *GGPPs7* from *Synechococcus sp*. and *crtE* from *X. dendrorhous*. These two GGPP syntheses showed good results increasing the precursor GGPP, which will supply the chain towards the β-carotene formation and further carotenoids and, therefore, were chosen to be used in this study [23,24]. The strain ST6899 was used as a parent strain to express and compare the activity of GGPPs7 and CrtE. It is worth noting that the parent strain already bore one copy of *crtE* from *X. dendrorhous*, one additional copy of the native *HMG1*, and the genes *crtI* and *crtYB* from *X. dendrorhous*. Additionally, the squalene synthase *SQS1* was downregulated by the shortening of the native promoter to 50 base pairs. The results showed that the strain carrying the GGPP synthase encoded by *GGPPs7* (ST7434) produced a β-carotene titer 272% higher (330 ± 66 mg/L) compared to the parent strain (88 ± 11 mg/L), whereas the strain ST7433, containing the second copy of *crtE* showed an increase of 48.59% compared with the parent strain, with a β-carotene production of 132 ± 11 mg/L (Figure 1).

### 2.2. Expression of Heterologous β-ketolases for the Biosynthesis of Astaxanthin Intermediates

The strain ST7434 with the highest titer of β-carotene was further engineered by insertion of β-ketolase coding genes from *Paracoccus* sp. (*PscrtW*) or from *H. pluvialis* (*HpBKT*), resulting in corresponding strains ST7906 and ST7972. Both strains had an orange-red color due to the formation of echinenone and canthaxanthin, which are β-carotene derivatives with one or two ketone groups, respectively. When analyzing the pathway, the keto-carotenoid canthaxanthin is the intermediate closer to astaxanthin and, therefore, a strain that accumulates higher amounts of canthaxanthin would be more relevant since it would have more potential to convert the high amount of canthaxanthin into astaxanthin. The HPLC analysis showed that the strain ST7972, which contained *HpBKT*, produced a higher amount of canthaxanthin (30 ± 2 mg/L) when compared to ST7906 (13 ± 0.8 mg/L). The strain ST7972 also produced 55 ± 3 mg/L of echinenone, and 82 ± 14 mg/L of β-carotene, while ST7906 accumulated 109 ± 5 mg/L of echinenone, and 30 ± 1 mg/L of β-carotene (Figure 2). The β-ketolases PsCrtW and HpBKT have a bifunctional activity and convert β-carotene into echinenone and echinenone into canthaxanthin. The higher titer of echinenone produced by ST7906 compared to ST7972 might suggest the preference of PsCtrW for β-carotene as substrate compared to echinenone, leading to an accumulation of this intermediate.

### 2.3. Single-Copy Expression of β-Hydroxylase for Production of Astaxanthin

To evaluate the production of astaxanthin in *Y. lipolytica*, the strains ST7906 and ST7972 were used as platforms for insertion of β-hydroxylases from the bacteria *P. ananatis* (*PaCrtZ*) or the microalgae *H. pluvialis* (*HpCrtZ*). Two to seven individual clones of each of the resulting four strains were screened for carotenoid production. We observed a significant clonal variation, which could possibility be due to the instability of the integrated β-ketolase genes, which were integrated into rDNA regions of the genome. Nevertheless, there was a clear tendency that strain ST7974, combining β-ketolase and β-hydroxylase genes from *H. pluvialis*, had the highest titer of astaxanthin (Figure 3; Figure 4), up to 20 ± 0.8 mg/L. This strain still produced significant amounts of astaxanthin precursors, 40 ± 2 mg/L of β-carotene, 47 ± 3 mg/L of echinenone, and 3 ± 0.5 mg/L of canthaxanthin.

### 2.4. Integration of Multiple Copies of β-ketolase and β-hydroxylase Increases Astaxanthin Production

The high concentrations of astaxanthin precursors (β-carotene, echinenone, and canthaxanthin) indicated imbalances in the pathway. In attempt to resolve these, we transformed a β-carotene producing strain ST7434 with different ratios of integration constructs targeting rDNA loci. For each transformation, two DNA constructs were mixed, one carrying a β-ketolase expression cassette and another carrying a β-hydroxylase expression cassette. As a positive control, we used the astaxanthin producer strain (ST7400) from our previous study [23]. Nine to seventeen individual clones were analysed for each of the four gene combinations (Figure 5; Figure 6). The highest titers of astaxanthin were again obtained for the combination of the genes from *H. pluvialis*; strain ST7976 isolate 3 gave astaxanthin titer of 44 ± 1 mg/L, which was 2.8-fold higher than the previously reported strain ST7400 with a titer of 15 ± 0.8 mg/L (Figure 6). Moreover, ST7976 (iso 3) accumulated 163 ± 12 mg/L of β-carotene. The carotenoid production details for all the strains constructed in this study can be found in Appendix A.

### 2.5. Fed-batch Fermentation of Astaxanthin Producer Strain

The production of astaxanthin by strain ST7976 (iso 3) was evaluated in 1 L controlled bioreactors. Fed-batch fermentation was performed on rich complex media with 20 g/L yeast extract, 40 g/L peptone, and with glucose. Glucose was added at a low rate in order to maintain its concentration below 5 g/L (Figure 7). After 2 days, 56 g DCW/L was accumulated and the growth stopped. The carotenoids were accumulated linearly from 24 h of fermentation until the end at 168 h. At the end, the astaxanthin titer reached 285 ± 19 mg/L (6 mg/g DCW) with the simultaneous production of 269 ± 44 mg/L of β-carotene, 42 ± 4 mg/L of echinenone, and 7 ± 0.8 mg/L of canthaxanthin (Figure 8). At the end of the process, 47% of the total carotenoids was astaxanthin. HPLC profile of the carotenoids produced by ST7976 (iso 3) at the end of the fementation process (168 h) can be seen in Appendix A. The high β-carotene titer shows the potential for further strain optimization so that all β-carotene can be converted into astaxanthin. The fermentation results show the potential of *Y. lipolytica* for production of astaxanthin.

## 3. Discussion

Owing to its outstanding antioxidant properties, health-related functions and application in the aquaculture and poultry sector, astaxanthin has a crescent market demand that is valued to reach USD 814 million by 2022 [25]. To meet this demand, astaxanthin production has been investigated in different microbial hosts. The strategies to improve the titer of astaxanthin varies from optimization of the astaxanthin biosynthetic pathway in native producers such as *X. dendrorhous* and *H. pluvialis* and using random mutagenesis to insertion of heterologous genes for astaxanthin production in new hosts such as *E. coli*, *S. cerevisiae*, and *Candida utilis* [7,18,26,27,28,29]. In this study, the oleaginous yeast *Y. lipolytica* was engineered for the production of astaxanthin. First, we evaluated the effect of two different GGPP synthases for the biosynthesis of β-carotene. The GGPP syntheses can come from a variety of sources, such as bacteria, fungi, or mammals. In this study, we compared the activity of the GGPP synthases CrtE from *X. dendrorhous* and GGPPs7 from *Synechococcus sp*. The results obtained in this study show that the expression of GGPPs7 increased, more efficiently, the carbon flux toward the formation of precursors to supply the astaxanthin production. The activity of GGPPs7 has been reported to be high enough to confer toxicity to the cell due to a dramatic increase in GGPP production, which could result in a drain on a downstream pathway such as ergosterol production [24]. Additionally, the expression of GGPPs7 for production of terpenoids has already been described in a patent by Evolva SA [30]. Several studies have successfully reported an increase in the production of isoprenoids by expressing native or heterologous *crtE* [18,23,31,32,33], while, to our knowledge, only a few patents have reported the expression of GGPPs7 for production of isoprenoids [24,30]. The results presented in this study show the potential of expressing GGPPs7 in *Y. lipolytica* to obtain higher titers of β-carotene.

Next, we analysed the efficiency of two different β-ketolases. The β-ketolase PsCtrW from *Paracoccus sp*. produced higher amounts of echinenone, an astaxanthin intermediate, compared to HpBKT from *H. pluvialis*. These data suggest the preference of this bifunctional enzyme (PsCrtW) for β-carotene as the substrate over echinenone. Then we integrated two different β-hydroxylases from either bacterial or microalgae organisms, into both platforms expressing PsCrtW and HpBKT. The gene expression was optimized by varying the copy number of the integrated genes. The best production of astaxanthin was obtained by the strain expressing the microalgae genes *HpBKT* and *HpcrtZ* in a molar ratio of 1:1. Although the transformation was performed in a way to balance the molar ratio between the two genes, the copy number of those need to be further investigated to evaluate if there is any disparity between the copy number of the enzymes. It is worth noting that the β-carotene hydroxylase adds a hydroxyl group to the β-carotene molecule, while the β-carotene ketolase adds a keto group. These two enzymes accept several substrates; thus, the β-carotene hydroxylase is capable of converting β-carotene to zeaxanthin and also canthaxanthin to astaxanthin. On the other hand, the β-ketolase can convert ß-carotene to canthaxanthin and also zeaxanthin to astaxanthin. Therefore, the high asxantathin content obtained in this study is a result of the enzymatic activity of HpcrtZ and HpBKT from *H. pluvialis*. In this study was used as positive control the astaxanthin producer strain of *Y. lipolytica* (ST7403) described in the work of Kildegaard et al. [23]. This strain of *Y. lipolytica* was engineered to produce astaxanthin and showed a production of 54.6 mg/L of astaxanthin. When compared to the positive control, the best astaxanthin producer obtained in this study presented a 145% higher titer in 24-well plates. This shows that the engineering modifications performed to improve the MVA pathway in order to increase the precursor supply plus overexpression of β-ketolase and β-hydrolase with higher activity has successfully increased astaxanthin production. Kildegaard et al. (2017) reported the production of astaxanthin using the bacterial genes *PscrtW* and *PacrtZ*, the results showed that the β-hydrolase PaCrtZ was the rate-limiting enzyme. Likewise, the present work has shown that when the bacterial genes are expressed together the best production of astaxanthin is achieved with a molar ratio of 1:3 (*PscrtW*: *PacrtZ*), confirming the β-hydrolase to be a rate-limiting step when expressing bacterial genes.

Finally, the best astaxanthin producer was cultivated in 1 L bioreactors and achieved a production of 285 ± 19 mg/L (6 mg/g DCW) of astaxanthin after 168 h of fermentation. The strain also produced 269 ± 44 mg/L of β-carotene, 42 ± 4 mg/L of echinenone, and 7 ± 0.8 mg/L of canthaxanthin. Previous studies have used the genes *BKT* and *HpCrtZ* from *H. pluvialis* to produce astaxanthin in *S. cerevisiae*, their results showed that when these enzymes were expressed, the astaxanthin stereoisomer obtained was the optically pure 3S, 3′S as the one produced in the microalgae *H. pluvialis* [18,19]. The chiral analysis for the astaxanthin produced in this present study was not performed, however, as the enzymes, BKT and HpCrtZ from *H. pluvialis* commonly produce the 3S, 3′S configuration, we believe that the same isomer is synthezed in *Y. lipolytica*. Nonetheless, future analysis to identify the astaxanthin stereoisomer synthetized by ST7976 (iso3) by NMR analysis, for example, is necessary. The high concentration of β-carotene in the strain ST7976 (iso 3) shows that further improvements in the pathway can lead to even higher titers of astaxanthin so that all β-carotene can be converted into astaxanthin. As demonstrated by Zhou et al. (2017) protein optimization of *BKT* from *H. pluvialis* led to higher activity of the enzyme and consequently higher production of astaxanthin [18]. Similarly, strategies to optimize rate-limiting enzymes in the pathways leading to biosynthesis of astaxanthin might be efficient to improve the production of astaxanthin. The fermentation process was carried in glucose limitation regime, since studies show that a high C:N ratio promotes lipid synthesis as well as the synthesis of carbon-based compounds, such as carotenoids [34]. Studies done by Larroude et al. 2017 showed that when glucose was kept at low concentration during the fermentation process, β-carotene production steadily increased [35]. In another study performed by Gao et al. 2017, β-carotene production also increased when glucose was at a low concentration in the medium [36]. Therefore, we selected a limited glucose regime for the fermentation process. Our results show that after nitrogen was exhausted from the medium, the biomass growth stopped, and a continuous increase in carotenoid production was observed. As described by Papanikolaou and Aggelis (2011), lipid production (secondary metabolite) in oleaginous yeasts is only triggered when a growth-required nutrient, in many cases nitrogen, is limited, and the carbon source is still available in the medium [37]. Similarly, our results suggest that a high C:N ratio positively affects astaxanthin production, which is also a secondary metabolite. Additionally, other strategies to improve the fermentation process and increase the astaxanthin biosynthesis could be applied, such as medium supplementation with Fe^2+^. In the work done by Zhou et al. (2017), the results showed that when Fe^2+^ was added in the media, the astaxanthin titer increased from 6.95 mg/g DCW to 8.10 mg/g DCW [18]. In another work, an increase in astaxanthin yield of 1.9-fold was reported when iron was supplemented into the media [19]. The positive effect of iron in the activity of the β-ketolase is associated to the histidine motifs present in the protein structure, which are reported to be involved in iron binding. The work done by Ye et al. (2016) reported that mutations in the conserved histidine motifs led to the inefficiency of the enzyme to catalyze the formation of ketocarotenoids [38]. Due to the time limitation, we had to stop the fermentation process at 168 h. Futher improvement of the fermentation process is required to achieve even better titer, rates, and yields. In addition, as the fermentation results showed a constant increase in astaxanthin production throughout the whole process, a longer fermentation might reveal even higher titers. To our knowledge only Kildegaard et al. (2017) reported the production of astaxanthin by engineered *Y. lipolytica* [23]. The obtained titer was 54.6 mg/L in microtiter plate cultivation, and fermentation was not performed. The results obtained in this work highlight the potential of *Y. lipolytica* for commercial production of astaxanthin.

## 4. Materials and Methods

### 4.1. Strains and Culture Conditions

*E. coli* DH5-α was used for the cloning procedures. The transformed *E. coli* cells were grown at 37 °C and 300 rpm in lysogeny broth (LB) liquid medium and at 37 °C on LB solid medium plates supplemented with 20 g/L agar. Ampicillin was supplemented when necessary at a concentration of 100 mg/L. The *Y. lipolytica* strain ST6899, engineered in previous work of Kildegaard et al. (2017) [23], was used as the parent strain. All strains used in this study are listed in Appendix A. *Y. lipolytica* was grown at 30 °C on yeast extract peptone dextrose (YPD), or synthetic complete minus Uracil (SC-Ura) media supplemented with 20 g/L agar. Supplementation with antibiotics was done when required at the following concentrations: hygromycin B at 50 mg/L and nourseothricin at 250 mg/L. The recombinant strains for carotenoids production were cultivated in yeast extract peptone medium containing 80 g/L glucose (YP + 8% glucose). The chemicals were purchased from Sigma-Aldrich, withexception of nourseothricin, which was purchased from Jena Bioscience GmbH (Jena, Germany).

### 4.2. Plasmid Construction

The geranylgeranyl pyrophosphate synthase encoded by *GGPPs7* from *Synechococcus* sp., the β-carotene ketolases encoded by *crtW* and *BKT* from *Paracoccus sp*. and *H. pluvialis*, respectively, and the β-carotene hydroxylases encoded by *crtZ* from *Pantoea ananatis* and *H. pluvialis* were codon-optimized for *Y. lipolytica* and synthesized as GeneArt String DNA fragments by Thermo Fisher Scientific (Waltham, MA, USA). The geranylgeranyl diphosphate synthase CrtE encoded by *crtE* from *X. dendrorhous* was obtained from Addgene [31]. The plasmids, primers, and BioBricks used in this study can be found in the Appendix A, respectively. The BioBricks were amplified by PCR using Phusion U polymerase (Thermo Fisher Scientific, Waltham, MA, USA) following the described conditions: 98 °C for 30 s; 6 cycles of 98 °C for 10 s, 51 °C for 20 s, and 72 °C for 30 s/kb; and 26 cycles of 98 °C for 10 s, 58 °C for 20 s, 72 °C for 30 s/kb, and 72 °C for 5 min. The BioBricks were purified from 1% agarose gel using the NucleoSpin^®^ Gel and PCR Clean-up kit (Macherey-Nagel, Bethlehem, PA, USA). After purification, the BioBricks were assembled into EasyCloneYALI vectors using USER cloning as described in the protocol by Holkenbrink et al. [39]. The USER reactions containing the desired plasmids were transformed into chemically competent *E. coli* DH5. The correct assembly was confirmed by DNA sequencing. Figure 9 summarizes the engineered pathway for improvement of precursor production and the expression of heterologous enzymes for the production of astaxanthin.

### 4.3. Construction and Cultivation of Y. lipolytica

Different previously characterized intergenic loci in *Y. lipolytica* were used to integrate yeast vectors into the genome of the parent strain, as described in Holkenbrink et al. [39]. To perform DNA transformation into *Y. lipolytica*, the integrative vectors were linearized with FastDigest NotI (Thermo Fisher Scientific, Waltham, MA, USA) and transformed into *Y. lipolytica* using a lithium-acetate protocol [40]. The transformants were selected on YPD + Hygromycin/Nourseothricin or SC-Ura plates. The yeast transformants carrying the correct integration in the genome were verified by colony PCR using primers listed in the Appendix A. The best β-carotenoid precursor producer was used for further implementation of the astaxanthin biosynthetic pathway. For astaxanthin production, the plasmids were constructed for single and multiple integrations in the *Y. lipolytica* genome. For multiple integrations, the vectors carried two homologous regions targeting the ribosomal DNA (rDNA) elements in *Y. lipolytica*. The transformation for multiple integrations was performed using vectors in three different molar ratios, 1:1, 1:2, and 1:3 (*PsctrW:PacrtZ*, *PscrtW:HpcrtZ*, *HpBKT:PacrtZ*, and *HpBKT:HpcrtZ*). The strain construction strategy is summarized in Figure 10. After the screening of transformants, 4 to 7 clones of astaxanthin producing strains were selected for each transformation molar ratio. The selection of colonies was based in color screening. For preculture preparation, single colonies were inoculated from fresh plates in 3 mL YPD in 24-well plates with an air-penetrable lid (EnzyScreen, Heemstede, The Netherlands). The strains were grown at 30 °C for 18 h with agitation of 300 rpm at 5 cm orbit cast. The required volume of inoculum was transferred to 3 mL YP + 8% glucose into 24-well plates for an initial OD600 of 0.1. The cultivation plates were incubated for 72 h at 30 °C with 300 rpm agitation. After cultivation, 0.5 mL of the cultivation volume was transferred into a prelabelled 2 mL microtube (Sarstedt, Numbrecht, Germany) for β-carotenoid extraction and subsequently quantification of carotenoids was done by HPLC.

### 4.4. Carotenoid Extraction

The optical density at 600 nm (OD600) was measured after cultivation, using NanoPhotometer (Implen GmbH, Munchen, Germany). For biomass dry weight measurements, 1 mL of the cultivation broth was transferred into a preweighed 2 mL microtube (Sarstedt, Numbrecht, Germany). The tubes were centrifuged at 10,000× *g* for 5 min. The supernatant was removed and the samples were washed with 1 mL of sterile water. Subsequent to the centrifugation and removal of the supernatant, the tubes containing the biomass pellets were placed in the incubator at 60 °C for 96 h. After 96 h the tubes were weighed on an analytical scale. For carotenoids extraction, 0.5 mL of the cultivation volume was transferred into a 2 mL microtube (Sarstedt, Numbrecht, Germany). Each sample was centrifuged at 10,000× *g* for 5 min and the supernatant was removed. Then, 0.5 mL of 0.5–0.75 mm acid-washed glass beads were added to each tube followed by the addition of 0.5 mL of ethyl acetate supplemented with 0.01% 3,5-di-tert-4- butylhydroxytoluene (BHT). The BHT was added to prevent carotenoid oxidation. The cells were disrupted using a Precellys R 24 homogenizer (Bertin Corp., Montigny-le-Bretonneux, France) in four cycles of 5500 rpm for 20 s. The tubes were placed on ice for 1 min in between each lysis cycle. After disruption, the cells were centrifuged for 5 min at 10,000× *g*. For quantification of β-carotene and individual carotenoids by HPLC, 100 µL of the solvent fraction was transferred to HPLC vials.

### 4.5. Carotenoid Quantification by HPLC

For HPLC measurements, 100 µL of ethyl acetate extract was evaporated in a rotatory evaporator, and the dry extracts were redissolved in 1 mL 99% ethanol + 0.01% BHT. Then, the extracts were analyzed by HPLC (Thermo Fisher Scientific, Waltham, MA, USA ) equipped with a Discovery HS F5 150 mm × 2.1 mm column (particle size 3 mm). For this analysis, the column oven temperature was set to 30 °C. All organic solvents used were HPLC grade (Sigma Aldrich, St. Louis, MO, USA). The flow rate was set to 0.7 mL/min with an initial solvent composition of 10 mM ammonium formate (pH = 3, adjusted with formic acid) (solvent A) and acetonitrile (solvent B) (3:1) until minute 2.0. Solvent composition was then changed at minute 4.0 following a linear gradient until % A = 10.0 and % B = 90.0. The solvent composition was kept until 10.5 min when the solvent was returned to initial conditions and the column was re-equilibrated until 13.5 min. The injection volume was 10 µL. The peaks obtained from the sample analysis were identified by comparison to prepared standards and integration of the peak areas was used to quantify carotenoids from obtained standard curves. The β-carotene and echinenone compounds were detected at retention times of 7.6 min and 6.9 min, respectively, by measuring absorbance at 450 nm, while astaxanthin and canthaxanthin were detected by absorbance at 475 nm and retention times of 5.9 min and 6.4 min, respectively. The results were verified by comparing the samples with the standards. Standards were purchased from Sigma-Aldrich: β-carotene (C4582-5 mg), echinenone (73341-1MG), canthaxanthin (11775-1MG), and astaxanthin (SML0982-50MG).

### 4.6. Fermentation Procedures

For inoculum, the strain glycerol stock was inoculated in 25 mL of media containing 20 g/L of yeast extract, 40 g/L of peptone and 5 g/L of glucose and propagated at 30 °C with 250 rpm agitation for 24 h. The OD600 was measured and the volume required to start the fermentation with an initial OD600 of 1.5 was transferred to a 20 mL syringe and used as inoculum. The fermentation was performed as fed-batch cultivation in a 1 L bioreactor (Sartorius Stedim Biotech, Gottingen, Germany). The initial cultivation volume was 0.4 L, the medium contained 20 g/L of yeast extract, 40 g/L of peptone, and 0.5 mL/L Antifoam 204 (Sigma, St. Louis, MO, USA). The 50% glucose stock solution, sterilized by filtration, was used as carbon source. To begin the fermentation, glucose was added to the concentration of 5 g/L. The temperature was kept constant at 28 °C, aeration was set to 2 VVM, the agitation was set to 500–1000 rpm, pH was automatically maintained at 5.5 by addition of 5 M KOH and 5 M HCl. The dissolved oxygen was set to a minimum of 20%. The foaming was prevented by automatic addition of Antifoam 204 (Sigma, St. Louis, MO, USA). The feeding of 50% glucose solution was initiated 6 h after inoculation. The glucose concentration was maintained below 5 g/L during the whole fermentation process and the glucose flow rate was adjusted manually according to the cell growth (OD600). The bioreactors were sampled three times a day to measure biomass dry weight, glucose, and carotenoids. For glucose quantification, the sample was immediately centrifuged, and the supernatant was stored at −20 °C until HPLC analysis.

### 4.7. Biomass and Glucose Quantification in Bioreactors

The OD600 values were detected with UV-1800 Shimadzu UV spectrophotometer. For the dry cell weight (DCW), 3 mL of the fermentation broth was filtered through preweighed cellulose nitrate membranes (0.45 µm pore size, 47 mm circle) using a filtration unit with a vacuum pump. The filters were dried at 60 °C for 96 h and weighed on an analytical scale. For glucose quantification, 1 mL of cultivation broth was transferred into a 2 mL microtube (Sarstedt, Numbrecht, Germany). The tubes were centrifuged at 10,000× *g* for 5 min. The supernatant was removed, filtered, and used for quantification on HPLC. The analysis on HPLC analyzed 20 µL of the sample for 30 min using an Aminex HPX-87H ion exclusion column with a 5 mM H2SO4 flow of 0.6 mL/min. The column temperature was set to 30 °C, the reflective index was set at 45 °C, and the glucose was detected using a RI-101 Refractive Index Detector (Dionex, Sunnyvale, CA, USA).

## 5. Conclusions

In this work, β-carotene production in *Y. lipolytica* was optimized through expression of GGPP synthase from *Synechococcus sp.* and then turned into astaxanthin producer by integration of heterologous β-ketolase and β-hydroxylase genes. The optimal gene combination was when both genes were from microalgae *H. pluvialis*. Nearly 0.3 g/L of astaxanthin was produced by the optimized strain in fed-batch cultivation with cellular content of 6 mg/g DCW. These results reinforce the potential of *Y. lipolytica* for production of carotenoids, in particular, astaxanthin.

## Figures and Tables

**Figure 1 microorganisms-07-00472-f001:**
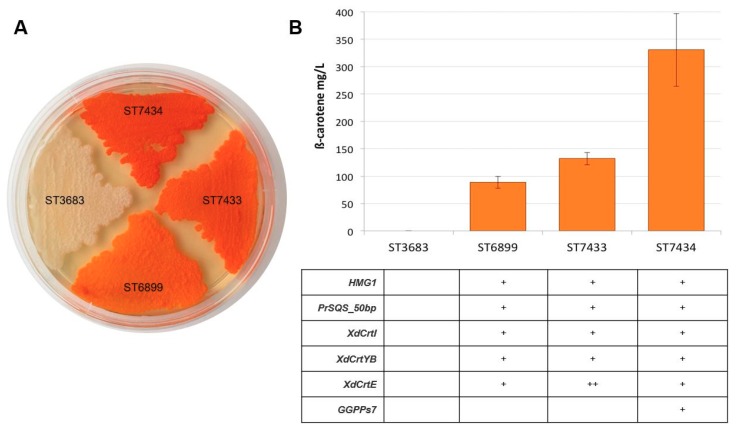
Effect of *Synechococcus GGPPs7* (ST7434) and the 2^nd^ copy of *crtE* (ST7433) on β-carotene production. ST6899 is the parental strain to ST7434 and ST7433. (**A**) Yeast extract peptone dextrose (YPD) plate after 2 days cultivation. (**B**) Titers measured by HPLC. The error bars represent standard deviations calculated from biological triplicate experiments.

**Figure 2 microorganisms-07-00472-f002:**
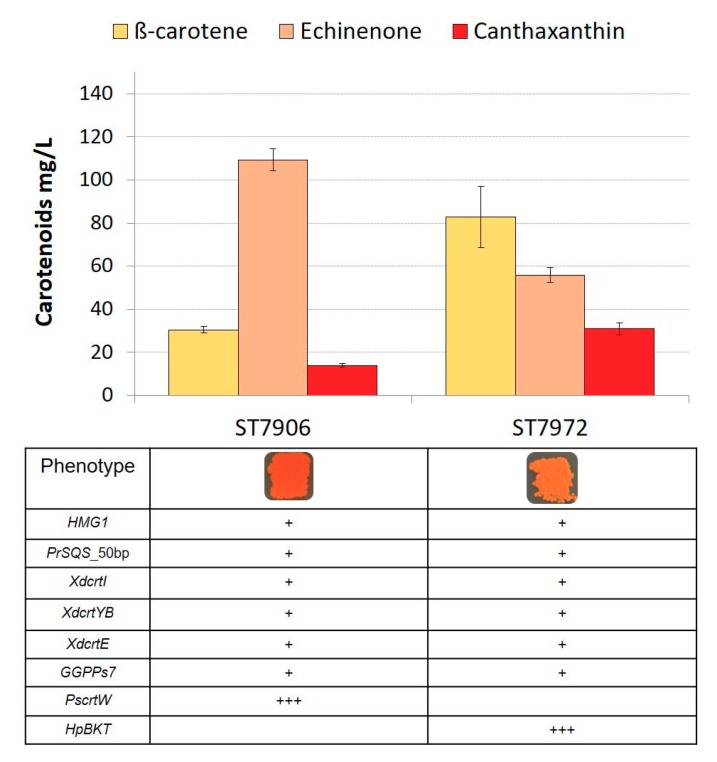
Carotenoid production of strains ST7906 (expressing multiple copies of *PscrtW*) and ST7972 (expressing multiple copies of *HpBKT*). All strains were cultivated in YP + 8% glucose in 24-deep-well plates for 72 h. The error bars represent standard deviations calculated from triplicate experiments.

**Figure 3 microorganisms-07-00472-f003:**
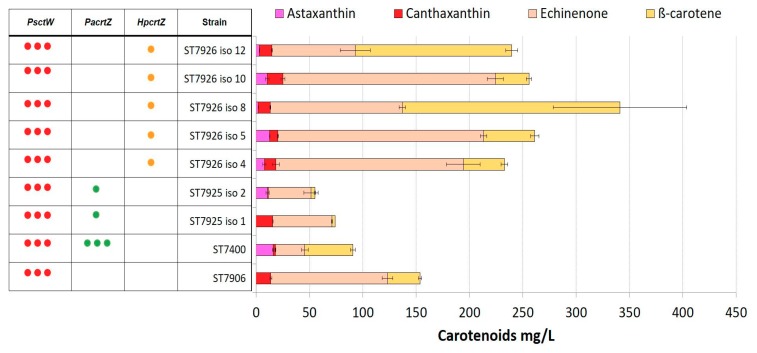
Carotenoid production by strains ST7925 and ST7926. Set of yeast transformants expressing *PscrtW* in combination with either *PacrtZ* or *HpcrtZ*. Positive control: ST7400; Parent strain: ST7906. All strains were cultivated in YP + 8% glucose in 24-deep-well plates for 72 h. Three dots represent multiple integrations of genes and one dot represents single integration. The error bars represent standard deviations calculated from triplicate experiments (“iso” after each strain indicates the isolate number).

**Figure 4 microorganisms-07-00472-f004:**
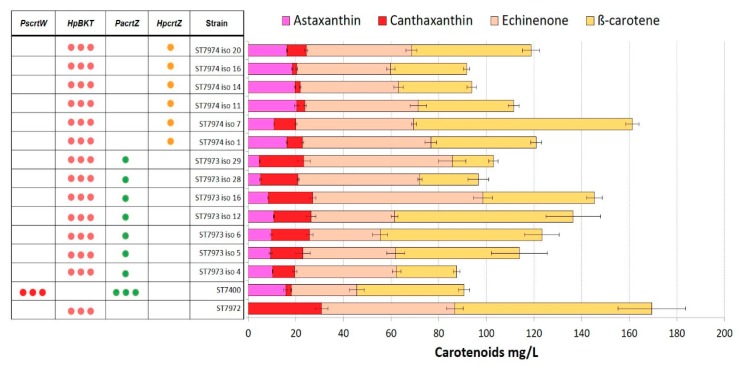
Carotenoid production by strains ST7973 and ST7974. Set of transformants expressing *HpBKT* in combination with either *PacrtZ* or *HpcrtZ*. Positive control: ST7400. Parent strain: ST7972. All strains were cultivated in YP + 8% glucose in 24-deep-well plates for 72 h. Three dots represent multiple integrations of genes, and one dot represents single integration. The error bars represent standard deviations calculated from triplicate experiments (“iso” after each strain indicates the isolate number).

**Figure 5 microorganisms-07-00472-f005:**
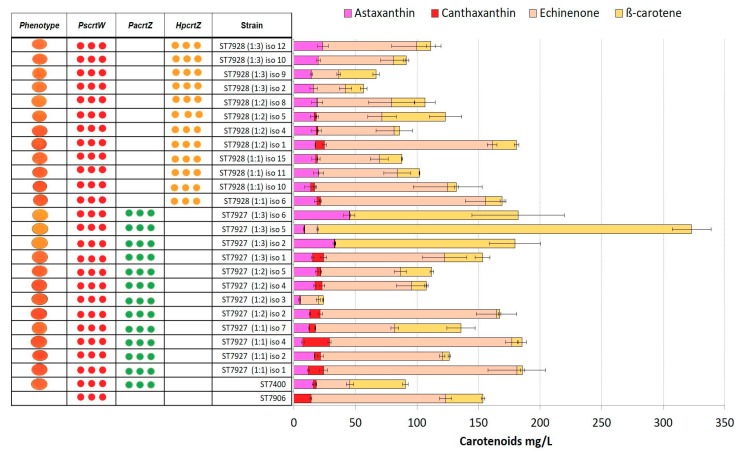
Carotenoid production by strains ST7927 and ST7928 transformed with combinations of *PscrtW* and *PacrtZ/HpcrtZ* genes. Molar ratios of DNA constructs are in brackets. Strains used as controls: ST7906 and ST7400. Three dots represent multiple integrations of genes. All strains were cultivated in YP + 8% glucose in 24-deep-well plates for 72 h. The error bars represent standard deviations calculated from triplicate experiments (‘iso’ after each strain indicates the isolate number).

**Figure 6 microorganisms-07-00472-f006:**
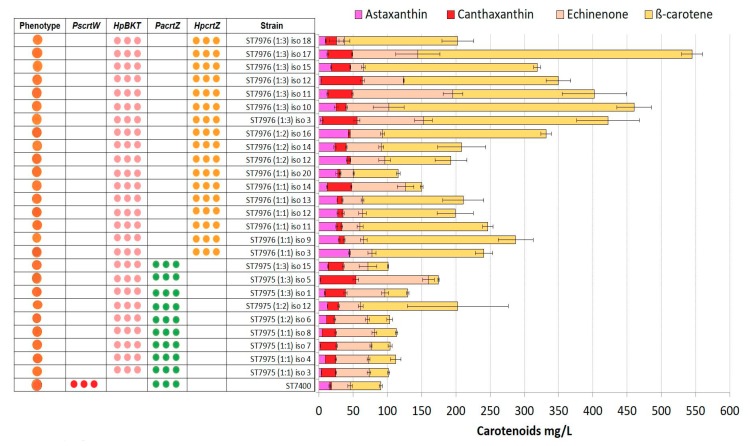
Carotenoid production by strains ST7975 and ST7976 transformed with combinations of *HpBKT* and *PacrtZ/HpcrtZ* genes. Molar ratios of DNA constructs are in brackets. Strain used as positive control: ST7400. Three dots represent multiple integrations of genes. All strains were cultivated in YP + 8% glucose in 24-deep-well plates for 72 h. The error bars represent standard deviations calculated from triplicate experiments (“iso” after each strain indicates the isolate number).

**Figure 7 microorganisms-07-00472-f007:**
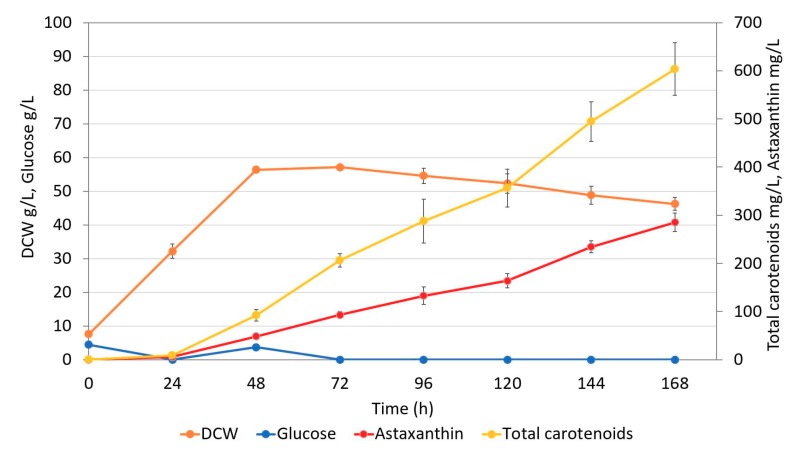
Concentrations of dry cell weight (DCW), glucose, and carotenoids during the fed-batch cultivations of ST7976 (iso 3). The values are averages from three independent experiments; the error bars represent show standard deviations.

**Figure 8 microorganisms-07-00472-f008:**
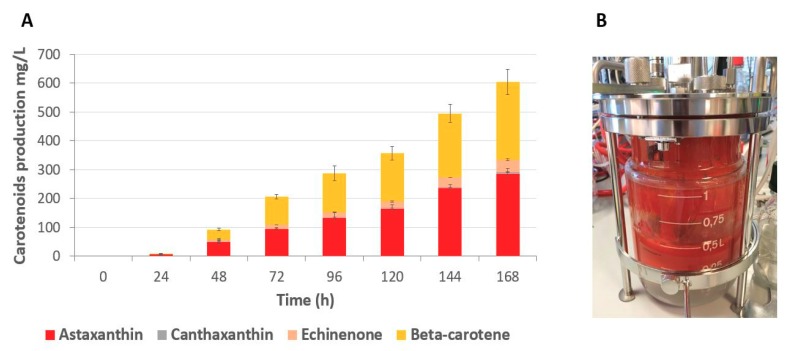
(**A**) Carotenoid concentrations during fed-batch cultivations (as in Figure 7) of ST7976 (iso 3). The values are averages from three independent experiments; the error bars represent show standard deviations. (**B**) Bioreactor at the end of the fed-batch fermentation of ST7976 (iso 3).

**Figure 9 microorganisms-07-00472-f009:**
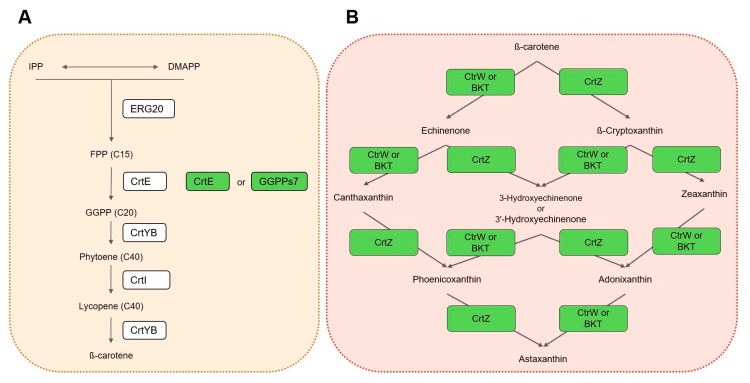
Engineered pathways. (**A**) Improvement of precursor supply. (**B**) Astaxanthin biosynthesis pathway. The white boxes indicate enzymes already expressed in the parental strain ST6899 [23]. The green boxes indicate enzymes additionally expressed in the parental strain in this study. IPP: Isopentenyl pyrophosphate; DMAPP: Dimethylallyl pyrophosphate; FPP: Farnesyl pyrophosphate; GGPP: geranylgeranyl pyrophosphate; ERG20: farnesyl pyrophosphate synthase; CrtE and GGPPs7: geranylgeranyl pyrophosphate synthase; CrtYB: phytoene synthase and lycopene cyclase; CrtI: phytoene desaturase; CrtW: β-ketolase from bacteria; BKT: β-ketolase from microalgae; CrtZ: β-hydroxylase.

**Figure 10 microorganisms-07-00472-f010:**
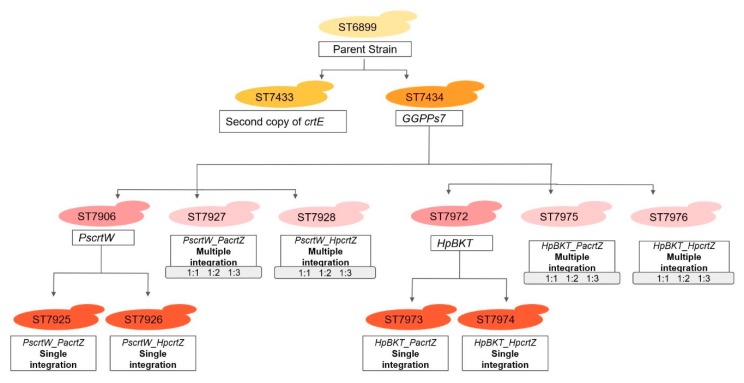
Flowchart of the strains generated in this study.

**Table 1 microorganisms-07-00472-t001:** Summary of astaxanthin production by natural producers and engineered non-carotenogenic organisms. Single black arrow (↓) represents downregulation.

Organism	Genotype	Astaxanthin Titer and Content	Reference
*X. dendrorhous*	*crtYB* and *asy* (native genes)	9.7 mg/g DCW (bioreactor)	[10]
*X. dendrorhous*	*crtYB*, *asy*, *crtE* and *trHMG* (native genes)	9 mg/g DCW (shake-flasks)	[11]
*H. pluvialis*	site-directed mutagenesis of *PDS* (native gene)	11.4 mg/g DCW (shake-flasks)	[12]
*E. coli*	*crtE*, *crtY*, *crtI*, *crtB*, *crtZ* (from *P. ananatis*); *trBKT* (from *C. reinhardtii*); *ispD* and *ispF* (native genes)	432 mg/L, 7 mg/g DCW (bioreactor)	[15]
*E. coli*	Module 1: *atoB* (native), *hmgS* (*S. cerevisiae*), *and thmgR (S. cerevisiae)*; module 2: *mevk* (*S. cerevisiae*), *pmk* (*S. cerevisiae*), *pmd* (*S. cerevisiae*), and *idi* (native); module 3: *crtEBI* (amplified from pAC-LYC plasmid) and *ispA* (native); *crtY* (*P. ananatis*), *crtZ* (from *P. ananatis*), *crtW* (*Brevundimonas sp*.)	320 mg/L, 2 mg/g DCW (SFE)	[16]
*S. cerevisiae*	***c****rtW* (from *Brevundimonas vesicularis*),*crtZ* (from *Agrobacterium aurantiacum*), and mutagenesis of *CSS1*, *YBR012W-B* and *DAN4*	217.9 mg/L, 13.8 mg/g DCW (bioreactor)	[17]
*S. cerevisiae*	*crtE*, *crtI*, *crtYB* (from *X. dendrorhous*); *trHMG1* (native gene); *BKT* and *crtZ* (from *H. pluvialis*)	47 mg/L, 8 mg/g DCW (shake-flasks)	[18]
*S. cerevisiae*	*BKT* and *crtZ* (from *H. pluvialis*)	4.7 mg/g DCW (shake-flasks)	[19]
*Y. lipolytica*	*crtYB*, *crtI*, *crtE* (from *X. dendrorhous*); *HMG1* (native gene); ↓*SQS1*; *crtW* (from *Paracoccus sp*.) and *crtZ* (from *P. ananatis*)	54.6 mg/L, 3.5 mg/g DCW (microtiter plates)	[23]
*Y. lipolytica*	*GGPPs7* (from *Synechococcus sp.*), *HpBKT*, *HpcrtZ* (from *H. pluvialis)*	285 mg/L, 6 mg/g DCW (bioreactor)	This study

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
