# Peer review of "Enhancement of Astaxanthin Biosynthesis in Oleaginous Yeast Yarrowia lipolytica via Microalgal Pathway"

_microorganisms, 2019, doi:10.3390/microorganisms7100472_

Round 1

Reviewer 1 Report

The corrected version can be published without further edition

Reviewer 2 Report

The changes made by the authors significativly improved the manuscript that is now suitable for publication on Microorganisms

This manuscript is a resubmission of an earlier submission. The following is a list of the peer review reports and author responses from that submission.

Round 1

Reviewer 1 Report

Referees Comments

on the manuscript entitled "Enhancement of astaxanthin biosynthesis in oleaginous yeast Yarrowia lipolytica via microalgal pathway” for “Microorganisms”

Currently, yeast Yarrowia lipolytica is widely studied as the producers for the practically valuable of products of microbial synthesis such as organic acids, lipids, proteins, etc. The authors presented the data about astaxanthin biosynthesis by Y. lipolytica. The great advantage of the article is the high methodological level of work. The results are presented with a logical sequence, beautifully illustrated. Of course, this work is of a high interest and in the scope of “Microorganisms” and could be considered for publication in this journal. But on my opinion, the minor revision of the manuscript is required.

Page 3, lines 105-109 - Please to reformulate the purpose of the work in a traditional manner. Page 3, Table 1 – Please increase the width of the column so that the name of the strain is on one line.  Please place the table on one page. Please to correct the name of the microorganisms in italics during the text.

Reviewer 2 Report

This paper from Tramontin LRR et al. describes Yarrowia lipolytica engineering to produce the high-value red pigment astaxanthin at high titers in submerged fermentation. The richest natural source for astaxanthin is the green alga Haematococcus pluvialis, which produces the most valuable isomer of this compound (i.e. 3S, 3′S astaxanthin). Chemical synthesis, instead, leads to the production of three different stereoisomers, (3S, 3′S); (3R, 3′S), and (3R, 3′R) in the ratio of 1:2:1, which are not suitable as feed additive in aquaculture.

The manuscript is interesting and well written. The authors have been able to develop a yeast factory producing high amounts of astaxanthin, giving adequate description of the methodological approach. However, I wonder why they did not perform a complete chemical characterization of the compound, considering that it can have a number of isomers. It would be important to characterize the distribution of astaxanthin isomers, due to the selectivity in metabolism of the different stereoisomers.

Minor points:

1) The author should explain why it is crucial to maintain glucose concentration below 5 g/l

2) Are there any data about carotenoids accumulation after 168 hours?

3) Lines 238 to 240 should be moved to “Carotenoid quantification by HPLC” paragraph.

Reviewer 3 Report

In this article, the authors report on an oleaginous yeast Yarrowia lipolytica was engineered to produce astaxanthin at high titers in submerged fermentation. The platform strain produced high content B-carotene, which is the precursor of astaxanthin. Different copy numbers of genes encoding B-ketolase and B-hydroxylase were introduced into the platform strain. After screen and cultivation in controlled bioreactors, the titer of astaxanthin could reach 285 mg/L. The production of astaxanthin increased as the design of the metabolism pathway. Publication in Microorganisms is recommended provided the authors address the following concerns:

-1 The authors have introduced several foreign genes to yeast, which seems to increase the yield of astaxanthin.  However, the authors did not show direct evidence of whether these foreign genes express enzymes protein or enhance related enzymes’ activities. It is still unclear whether the effect comes from the introduction process of the gene or from the enzyme protein expressed by the gene.

-2 HPLC profiles of the different astaxanthin and its derivates should be included in a supporting information file. Ideally, the identity of them should be also confirmed by ES-MS or other analysis.

-3 Many names of species are not italic.